# Assessing behavioural profiles following neutral, positive and negative feedback

**Rimsa Dahal[1], Kelsey MacLellan[1], Danielle Vavrek[1], Benjamin James Dyson** [1,2,3] *

**1** University of Alberta, Edmonton, Canada, **2** University of Sussex, Brighton, United Kingdom, **3** Toronto Metropolian University, Toronto, Canada

* bjdyson@ualberta.ca

## Abstract

Previous data suggest zero-value, neutral outcomes (*draw*) are subjectively assigned negative rather than positive valence. The combined observations of faster rather than slower reaction times, subsequent actions defined by *shift* rather than *stay* behaviour, reduced flexibility, and, larger rather than smaller deviations from optimal performance following draws all align with the consequences of explicitly negative outcomes such as losses. We further tested the relationships between neutral, positive and negative outcomes by manipulating value salience and observing their behavioural profiles. Despite speeded reaction times and a non-significant bias towards *shift* behaviour similar to losses when *draws* were assigned the value of 0 (Experiment 1), the degree of *shift* behaviour approached an approximation of optimal performance when the draw value was explicitly positive (+1). This was in contrast to when the draw value was explicitly negative (-1), which led to a significant increase in the degree of *shift* behaviour (Experiment 2). Similar modifications were absent when the same value manipulations were applied to win or lose trials (Experiment 3). Rather than viewing draws as neutral and valence-free outcomes, the processing cascade generated by draws produces a complex behavioural profile containing elements found in response to both explicitly positive and explicitly negative results.

## 1. Introduction

Responsiveness to feedback is a fundamental aspect of learning (e.g., [1]). Principles of operant conditioning clearly express the common ways in which the outcome of current behaviour influences future action [2]. While changing action as a consequence of negative outcome (ie, *lose-shift*) and repeating action as a consequence of positive outcome (ie, *win-stay*) seem to be two sides of the same coin, punishment and reinforcement remain anatomically [3], evolutionarily [4, 5] and mechanistically [6, 7] distinct. In addition to operating in environments such as gambling and education- where feedback is both salient and explicitly negative or positive- there are other cases where information regarding our performance is often ambiguous or incomplete [8]. The interpretation of outcomes that do not have a clear valence, either as a result of the absence of feedback (ambiguous; [9]) or the explicit delivery of neutral feedback, such as the zero value assigned to drawing against an opponent [10], is a neglected feature of the decision-making literature.

**Data Availability Statement:** All summary data, raw data and scripts are available at PsyArXiv https://osf.io/9wnjq/.

**Funding:** The lab is funded by an NSERC Discovery Grant (RGPIN-2019-04954), Alberta Gambling

Research Institute Grants, and start-up monies provided by the University of Alberta (RES0042096). The funders had no role in study design, data collection and analysis, decision to publish, or preparation of the manuscript.

**Competing interests:** The authors have declared that no competing interests exist.

The behavioural and neural responses following supposedly 'neutral' outcomes have provided a number of unique insights into the subjective aspects of decision-making. Within a simple case where three possible outcomes of a competitive interaction are assigned different values (*win* = +1, *draw* = 0, *lose* = -1), a draw may co-opt aspects of positive or negative outcomes as a result of the transient state of the organism (see [11]). [5] (2001, p. 225) express this interpretive tension generated by draws as "to tie is to fail to win, but on the other hand to tie is to avoid a loss." Thus, a draw may be perceived as *worse-than-expected* in the context of not winning, or *better-than-expected* in the context of not losing. This need for personal interpretation means that individuals- in otherwise identical environments in which they encounter the same types and frequencies of outcomes- can form different subjective states where they experience more successes than failures (i.e., [win = draw] > lose), or, more failures than successes (i.e., win < [draw = lose]). This is important for at least two reasons. First, given the inherent ambiguity of draws, these responses can speak to the degree of optimistic bias or depressive realism held as a trait [12, 13]. Second, draws play a critical component in determining gambling behaviour, since a draw could be equally perceived as either a near-win or a near-loss thereby perpetuating erroneous beliefs in performance success [14, 15]. Given the ubiquity of traditional operant conditioning responses to clear gains and losses (*win-stay*, *lose-shift*), it seems probable that the behavioural profile for *draw* outcomes has its hallmark in these more hard-wired reactions. We can start to understand the subjective interpretation of draws by comparing performance with explicitly positive and negative outcomes, in addition to reviewing the previous literature in terms of a number of metrics including decision times, neural flexibility, and, the quality and optimality of action following outcome.

With respect to decision time, we can draw from the literature on *post-loss slowing / speeding* (c.f., impulsivity; [16–18]). Here, differences in future decision time are determined by the type of outcome caused by the current action: *post-loss slowing* is defined as increased decision time following losses relative to wins, and, *post-loss speeding* is defined as decreased decision time following losses relative to wins. A contributing factor in the observation of slowing or speeding is the degree to which failure is rare [19]. For example, if an individual interacts with an opponent who cannot be beaten (*unexploitable*), or interacts with an opponent who can be beaten (*exploitable*) but who the participant fails to beat, then performance is characterized by *post-loss speeding*. In contrast, the degree to which individuals successfully exploit an opponent increases the magnitude of *post-loss slowing* (see [20]; Fig 1). However, outcome frequency does not provide a complete account of post-loss slowing since post-loss slowing is intact when positive and negative outcomes are experienced to the same degree (eg., [21]). Previous data utilizing draw outcomes further show that when participants engage with *unexploitable* opponents where long-run outcome frequencies were equivalent, decision times following wins were slower than decision times following both losses and draws (i.e., *post-draw speeding*; [21, 22]). Therefore, decision time speeding data establish a connection between neutral (draw) outcomes and explicitly negative (lose) outcomes.

A second metric to ascertain the subjective interpretation of draws is the response of the brain following outcomes. Feedback-related negativity (FRN; [23]) is a scalp-recorded electrical potential related to dopaminergic influences on anterior cingulate cortex (ACC; [24]) and is a reliable index of the positive or negative nature of trial outcome [25]. Paradigms directly using neutral outcomes have shown that FRN amplitudes generated following draws are often statistically indistinguishable from unambiguously negative (i.e., lose) outcomes (e.g., [10, 25, 26]). In contrast, both lose and draw trials generate larger FRN amplitude than unambiguously positive trials (i.e., win; e.g., [27, 28]). Therefore, event-related potential data also align draws with a negative rather positive interpretation.

## EXPERIMENT 1

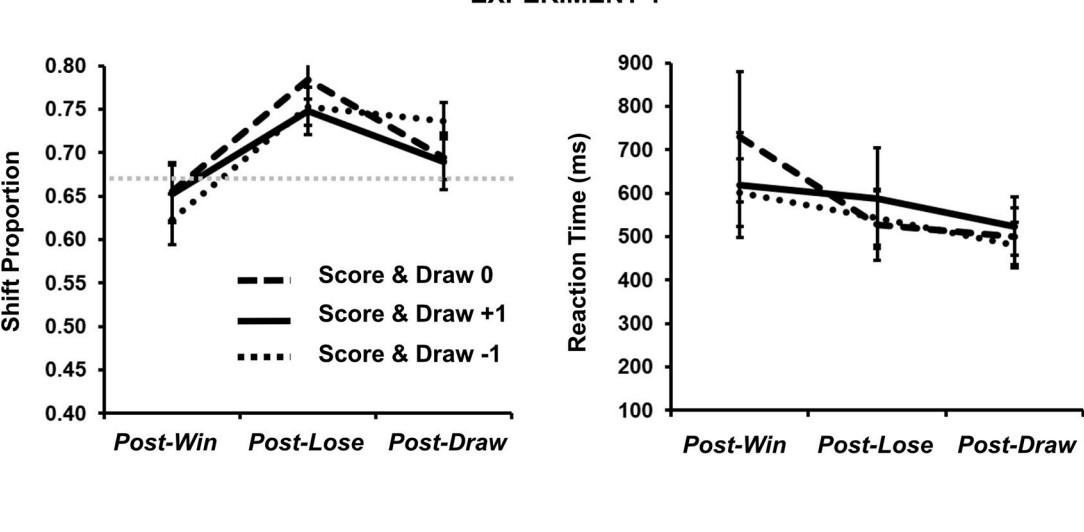

## EXPERIMENT 2

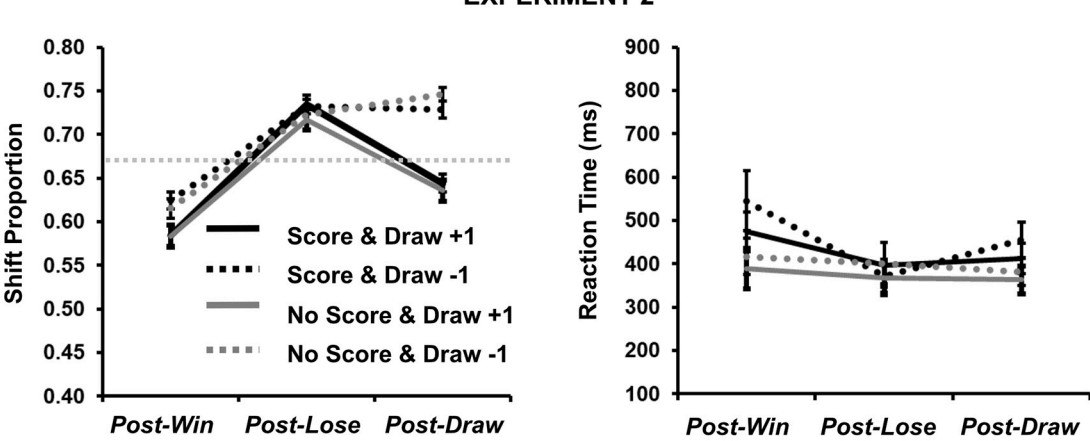

## EXPERIMENT 3

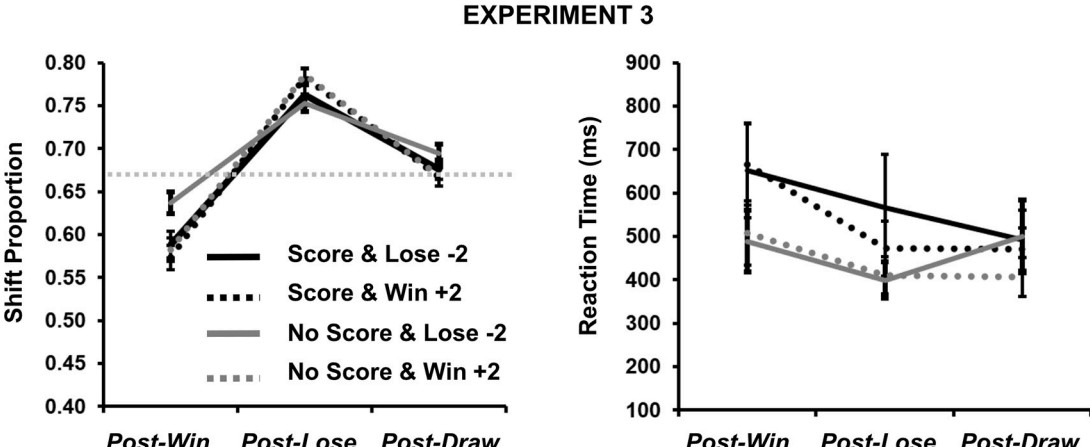

**Fig 1. Left column depicts proportion of *shift* behaviour following wins, losses and draws as a function of cumulative score presentation (score, no score) and the value of draws (0, +1, -1; Experiment 1), (+1, -1; Experiment 2), or, the value of losses (-2) and wins (+2; Experiment 3).** Horizontal grey dotted line represents optimal performance (66.6%). Right column depicts subsequent reaction time following wins, losses and draws. Error bars represent standard errors.

Of final interest is the flexibility of behaviour exhibited following outcome. Operant conditioning dictates the tendency to repeat action following success (*win-stay*) and the tendency to switch action following failure (*lose-shift*). Under baseline conditions (win = +1, draw = 0, loss = -1), the degree of *win-stay* behaviour approximates the optimal proportion (33.3%) as predicted by a mixed-strategy-style approach *guaranteeing* the absence of exploitation (see [10, 21; Experiment 1]). In contrast, the degree of *lose-shift* behaviour exceeds the optimal proportion (66.6%), making participants more predictable following loss. In follow-up studies [22], the objective values of losses and wins was manipulated relative to baseline (see also [6, 29]). Data produced by these schemes showed that value manipulations changed behaviour for wins but not for losses [6, 22]. Specifically, even when the objective cost of losing was reduced to half the gain of winning (-1, +2), participants still exhibited robust *lose-shift* behaviour to the same degree observed in a baseline condition (-1, +1) or a condition in which the objective cost of losing was doubled relative to the gain of winning (-2, +1). In contrast, *win-stay* behaviour was more flexible and significantly increased when wins and losses had different objective values. Therefore, negative outcomes are characterized by subsequently less flexible *shift* behaviour, whereas positive outcomes are characterized by subsequently more flexible *stay* behaviour. One of the basic principles of loss aversion is that losses have roughly twice the subjective magnitude of their objective value [22, 30, 31]. Therefore, the inflexibility of shift behaviour following loss might be associated with the greater subjective value of negative relative to positive outcomes.

From our previous data however, we note a seemingly reliable observation that prohibits the wholesale acceptance of *draws* as functionally equivalent to *losses*. For example, in [18] (Experiment 1, no credit condition), the degree of *shift* behaviour is smaller following *draws* relative to *losses* (76.63% vs. 70.62%; t[39] = -2.267, p = .029; *n* = 40). Similar reductions in *shift* behaviour following *draws* relative to *losses* were also extracted from [10] (Experiment 1: 72.53% vs. 78.51%; *t*[35] = -2.530, *p* = .016; *n* = 36), [22] (baseline condition; 71.97% vs. 77.68%; t[35] = -2.120, p = .041; *n* = 36), and, [21] (70.63% vs. 75.89%; t[30] = -2.307, p = .028; *n* = 31). Given the attenuation of *shift* bias following *draws* relative to *losses*, then there is clearly some sense in which these are not identical examples of negative outcomes. Draws generate a less negative state, which enables the individual to approach optimized performance in the long run (ie 66.6% *shift* behaviour).

## 2. Experiment 1

In Experiment 1, participants engaged in the three-response game Rock, Paper, Scissors using a novel objective value manipulation for draws (+1, 0, -1). Critically, performance was evaluated against a computerized opponent playing according to a mixed-strategy variant, where each of the three responses appeared 33.3% of the time, in a random order. (MS; [32–34]).

In principle, playing against an opponent operating according to MS should guarantee- in the long run- an equivalent number of wins, losses and draws. Thus, when the opponent is *unexploitable*, there is no reliable model of successful performance for participants to acquire. Once again, this type of performance is critical for the current debate as it allows the same exposure to positive (*win*), neutral (*draw*) and negative (*lose*) outcomes. Starting at the position that *draws* are objectively neutral (0), we consider the consequences of making the value outcome of *draws* equivalent to *wins* (+1) and *losses* (-1). If objective value is the only factor contributing to performance, then the behavioural changes we see when *draws* are assigned +1 and -1 should be of the same magnitude. If value and valence interact, then the behaviour following +1 and -1 draws should be different. Since *draws* are observed to have both behavioural

[21, 22] and neural [25, 27] hallmarks similar to objectively negative outcomes (*losses*), moving the value to +1 should be a more salient manipulation than moving the value to -1.

Changing draw value from 0 to +1 or -1, and evaluating performance against an *unexploitable* opponent in Experiment 1, we make the following predictions. This is guided by data from 12 previously published experiments that use a similar *win* (+1), *draw* (0), *lose* (-1) outcome value assignment against an opponent who cannot be beaten [10, 18, 20–22, 35]. First, reaction times following draws should be faster than reaction times following wins (*post-draw speeding*). Second, actions following draws should be more likely to elicit *shift* than *stay* behaviour, once again aligning with the consequences of losing rather than winning. Third, the degree to which participants stray from optimal performance should be greater for *draw-shift* than *win-shift*, consistent with the behavioural sub-optimality often associated with negative relative to positive outcomes. The expected value for all shift behaviour is 66.6%. Fourth, if draws are interpreted as negative when assigned the value of 0, changing the value of the draw to +1 (i.e., moving from subjectively negative to objectively positive) should alter behaviour more than changing the value of the draw to -1 (i.e., moving from subjectively negative to objectively negative).

## 2.1 Method

40 participants provided informed consent for Experiment 1 (37 provided demographic information: 28 women, 32 right-handed; mean age = 19.03, *sd* = 1.11). Participants received course credit and were only eligible to take part in one experiment in the following series. The protocol was approved at the University of Alberta under Research Ethics Board 2 (Pro00083768). Sample sizes were estimated with 80% power [36] using G*Power [37] from two recent studies described above, in which reduction in *shift* behaviour following *draw* (0) relative to *loss* (-1) trials were observed: [18]: $dz$ = 0.6894, one-tailed, yielding $n$ = 15, and, [10]: $dz$ = 0.4217, one-tailed, yielding $n$ = 37.

Static pictures of blue-gloved (left; opponent) and white-gloved (right; participant) hands signalling Rock, Paper and Scissors poses (from [22]) were displayed center screen at approximately 12˚ x 6˚, with participants sat approximately 57 cm away from a 22" ViewSonic VX2757 Monitor. Stimulus presentation was controlled by Presentation 20.2 (build 07.25.18) and responses were recorded using a keyboard.

Participants completed 3 counterbalanced blocks of 120 trials. Within each block, the computerized opponent played Rock, Paper, Scissors 40 times each in a random order. With wins consistently assigned the value of +1 and losses consistently assigned the value of -1, the only difference between the conditions was the value assigned to draw trials (*+1, 0, -1*). At the beginning of each block, participants were informed how much game outcomes were worth. At the start of each trial, the participant was presented with a fixation cross and cumulative score for the opponent (bottom left) and player (bottom right). After pressing 4, 5 or 6 on the number pad (representing Rock, Paper, Scissors), the choices made by the opponent and the participant were displayed for 1000 ms. Choices were cleared over a 500 ms period, after which the outcome of the trial was displayed for 1000 ms. Scoring was updated during 500 ms and the next trial began.

## 2.2 Results

Statistica 13.3 (TIBCO Software) was used to analyze the data. The comparison of *wins* (32.92%), *loses* (33.97%) and *draws* (33.12%) collapsed across condition was not significant according to a one-way repeated-measures ANOVA [$F(2,78)$ = 1.274, MSE < .001, $p$ = .286, $_{\text{p}}^{2}$ = .031], suggestive of long-run equivalence of outcomes in Experiment 1. Median RTs on trial

**Table 1. Distribution of item, outcome, outcome-response contingency and reaction time, and randomness deviation as a function of the value of draw trials (+1, 0, -1) in Experiment 1.** Standard error in parenthesis.

| | Item | | | Outcome | | |
|---|---|---|---|---|---|---|
| | Rock | Paper | Scissors | Win | Lose | Draw |
| D+1 | .370 (.020) | .323 (.013) | .307 (.011) | .327 (.006) | .336 (.008) | .337 (.007) |
| D0 | .356 (.013) | .327 (.011) | .316 (.011) | .322 (.006) | .347 (.007) | .331 (.007) |
| D-1 | .348 (.012) | .335 (.010) | .318 (.015) | .339 (.007) | .336 (.006) | .325 (.007) |
| | Outcome-Response Contingency | | | Reaction Time (ms) | | |
| | Win-Stay | Lose-Shift | Draw-Shift | Win | Lose | Draw |
| D+1 | .348 (.033) | .748 (.028) | .690 (.033) | 619 (121) | 588 (115) | 524 (67) |
| D0 | .345 (.033) | .784 (.023) | .695 (.026) | 729 (150) | 527 (82) | 501 (66) |
| D-1 | .377 (.029) | .753 (.022) | .736 (.021) | 600 (78) | 542 (62) | 480 (53) |

*n* following *wins*, *loses* and *draws* at trial *n-1* were compared across the three conditions (draw value: *+1, 0, -1*) in a two-way repeated-measures ANOVA (see right-hand panel of Fig 1). Only a significant main effect of outcome was revealed [$F(2,78) = 6.334$, MSE = 107065, $p = .003$, $_p^2 = .140$], with significant speeding for *draws* (502 ms) relative to *wins* (hence, *post-draw speeding*; 650 ms; Tukey's HSD, $p = .002$) but not between *losses* (553 ms) and *wins* ($p = .062$). The main effect of draw value [$F(2,78) = 0.327$, MSE = 207042, $p = .722$, $_p^2 = .008$], and, the interaction between draw value x outcome [$F(4,156) = 2.136$, MSE = 43307, $p = .003$, $_p^2 = .052$] were not significant.

The proportion of *shift* behaviour following wins, losses and draws was calculated from the last 119 trials in each block (the first trial has no history; see Table 1). In this way, an observed value less than the expected value of 66.6% represents a bias towards *stay* behaviour, where an observed value more than the expected value of 66.6% represents a bias towards *shift* behaviour. Shift proportions were compared across outcome (*win*, *lose*, *draw*) and draw value *(+1, 0,-1)* in a two-way repeated-measures ANOVA (see left-hand panel of Fig 1; the horizontal grey line represents the expected value of 66.6%). The main effect of draw value was not significant [$F(2,78) = 0.215$, MSE = .029, $p = .807$, $_p^2 = .005$], nor was the interaction between draw value x outcome [$F(4,156) = 2.420$, MSE = .010, $p = .051$, $_p^2 = .058$]. A significant main effect of outcome [$F(2,78) = 16.874$, MSE = .025, $p < .001$, $_p^2 = .302$] demonstrated that the degree of *shift* behaviour for wins (64.35%) was smaller than that for both losses (76.17%) and draws (70.70%). The degree of *draw-shift* behaviour was also significantly smaller than the degree of *lose-shift* behaviour (all Tukey's HSD, $p < .05$).

Observed *shift* proportions following wins, losses and draws were also compared to the expected value predicted by the participant playing according to MS (66.6%) via one-sampled t-tests. Neither *win-shift* ($t[39] = 0.875$, $p = .387$) nor *draw-shift* ($t[39] = 1.885$, $p = .067$) was significantly different from expected value, but *lose-shift* ($t[39] = 5.163$, $p < .001$) proportions did supporting a significant bias towards shift behaviour following losses.

## 2.3 Discussion

Experiment 1 establishes a number of key observations with respect to the subjective interpretation of draw outcomes. First, there was a clear indication of *post-draw speeding*, in alignment with negative rather than positive valence interpretation. Second, behaviour following *draws* was characterized by *shift* (70.70%) rather than *stay* (29.30%) behaviour, again consistent with reactions to negative (i.e., *lose-shift*) rather than positive (i.e., *win-stay*) outcomes. Third, in terms of the quality of behaviour as a consequence of outcome, *shift* proportions following

draws were closer to optimized MS performance compared to losses. This leads us away from the wholesale adoption of the view that draws are functionally equivalent to losses.

Finally, we note changing the value of a draw to -1 did not significantly impact behaviour less than changing the value of a draw to +1. Instead, an examination of Table 1 shows a non-significant increase of *shift* behaviour when draws were assigned -1 (73.6%), relative to when draws were assigned both 0 (69.5%) and +1 (69.0%).

## 3. Experiment 2

The data from Experiment 1 identified distinct behavioural effects that underline the ambiguities associated with draws. Draws were aligned with negative outcome in that future actions were characterized by *speeding* rather than *slowing*, and, *shift* rather than *stay* behaviour, similar to losses. However, draws did not generate *identical* states as losses, since the degree of shift behaviour was equivalent to MS performance in the former case. Draws were also weakly aligned with positive outcomes in that the degree of shift behaviour was constant when the value of the draw was made explicitly positive but rose (non-significantly) when the value of the draw was made explicitly negative. In Experiment 2, we focused more specifically on the hypothesis that shift behaviour should increase- away from MS performance- via the explicit assignment of draw value equivalent to losses (-1) compared to wins (+1; c.f., [38]).

We also considered the contribution of a potentially hidden variable in Experiment 1- namely, the presence of a cumulative score. In addition to the immediate effect of adding or subtracting individual points on a trial-by-trial basis, cumulative scores provide a longer-range index of success or failure. This subjective, longer-range evaluation of outcome has support at a neural level, given the observation that the anterior cingulate cortex (ACC) represents current outcomes against the broader context of average task value [39]. In particular, the presence of such cumulative scores may have been particularly important in Experiment 1 exactly due to the manipulation of draw value. Since MS opponents were used in all three conditions, participants experienced a broadly equivalent number of *wins*, *draws* and *losses*. Therefore, the degree of success / failure was equivalent across conditions. However, when draw outcomes were assigned to non-zero values, the use of +1 guaranteed an increasingly positive score as the block progressed, in contrast to the use of -1 guaranteeing an increasingly negative score. This is clearly borne out in the data from Experiment 1: the final average score when *draw* = 0 was -3.00 (*SE* = 1.30), when *draw* = +1 was +39.25 (*SE* = 1.85), and, when *draw* = -1 was -38.65 (*SE* = 1.72; F[2,78] = 628.15, MSE = 96.84, p < .001, $_p^2$ = .942; one-way repeated measures ANOVA, all comparisons, Tukey's HSD *p* < .05).

Therefore, it is possible that experiencing an increasingly positive or negative score impacts upon any immediate effects generated by individual trial-by-trial outcome values [39]: positive scoring (+1) become less salient against a backdrop of a reliably increasing total, just as negative scoring (-1) becomes less salient against a backdrop of a reliably decreasing total. Consequently, in Experiment 2, the central manipulation of draw value in accordance with explicitly positive (+1) or explicitly negative (-1) outcomes was combined with the manipulation of the presence or absence of a cumulative score.

### 3.1 Method

36 participants were analyzed for Experiment 2 (for the 35 individuals who provided demographic information: 26 women, 32 right-handed; mean age = 19.40, *sd* = 0.32). One individual was replaced due to experimenter error. Four conditions were completed in a counterbalanced order, in which the value of draw (+1, -1) and cumulative score (present, absent) varied. Blocks were now 90 trials (4 conditions) rather than 120 trials (3 conditions) used in Experiment 1.

**Table 2. Distribution of item, outcome, outcome-response contingency and reaction time as a function of the value of draw trials (+1, -1) and presence (S) or absence (NS) of cumulative score in Experiment 2.** Standard error in parenthesis.

| | | Item | | | Outcome | | |
|---|---|---|---|---|---|---|---|
| | | Rock | Paper | Scissors | Win | Lose | Draw |
| D+1 | S | .333 (.015) | .327 (.015) | .340 (.014) | .353 (.007) | .329 (.008) | .318 (.007) |
| D+1 | NS | .345 (.016) | .339 (.013) | .316 (.016) | .325 (.009) | .342 (.008) | .333 (.007) |
| D-1 | S | .333 (.013) | .324 (.011) | .343 (.011) | .331 (.008) | .345 (.008) | .324 (.007) |
| D-1 | NS | .338 (.013) | .342 (.012) | .319 (.011) | .326 (.008) | .335 (.009) | .339 (.009) |
| | | Outcome-Response Contingency | | | Reaction Time (ms) | | |
| | | Win-Stay | Lose-Shift | Draw-Shift | Win | Lose | Draw |
| D+1 | S | .415 (.039) | .734 (.035) | .644 (.033) | 474 (49) | 397 (55) | 412 (37) |
| D+1 | NS | .417 (.040) | .717 (.013) | .636 (.039) | 389 (50) | 367 (41) | 363 (34) |
| D-1 | S | .376 (.030) | .723 (.026) | .728 (.031) | 546 (72) | 371 (40) | 455 (43) |
| D-1 | NS | .385 (.036) | .722 (.027) | .746 (.024) | 417 (44) | 401 (51) | 381 (32) |

Participants also reported their subjective impression of each condition along a visual analog scale from total luck to total skill, as part of a larger empirical exercise to be reported elsewhere. All other parameters in Experiment 2 were identical to Experiment 1.

## 3.2 Results

As in Experiment 1, the comparison of *wins* (33.39%), *losses* (33.75%) and *draws* (32.85%) was not significant [$F(2,708) = 0.757$, MSE < .001, $p = .473$, $_p^2 = .021$; see Table 2]. Median trial $n$ RTs were submitted to a three-way repeated-measures ANOVA featuring draw value (+1, -1), cumulative score (present, absent) and outcome on trial *n-1* (*win*, *lose*, *draw*). Only the two-way interaction between cumulative score x outcome was significant: [$F(2,70) = 5.42$, MSE = 18945, $p = .006$, $_p^2 = .134$]. The interaction showed both *post-loss speeding* (384 ms) and *post-draw speeding* (434 ms) relative to RTs following wins (510 ms) but only in cases where the cumulative score was present (Tukey's HSD, $p$s < .05).

Reaction times when the cumulative score was present following wins were also slower than all outcomes when the cumulative score was absent (384, 372, 403 ms for losses, draws and wins, respectively). All other comparisons were non-significant, with the interaction arising from the magnitude of *post-loss* and *post-draw speeding* decreasing without cumulative score. In these respects, the presentation of an increasingly positive-going / negative-going score would appear to be a prerequisite for the reliable observation of *speeding following 'negative' outcomes*.

All other main effects and interactions were not significant: cumulative score main effect [$F(1,35) = 3.029$, MSE = 112624, $p = .091$, $_p^2 = .079$], draw value main effect [$F(1,35) = 1.347$, MSE = 63459, $p = .254$, $_p^2 = .037$], outcome main effect [$F(2,70) = 2.789$, MSE = 71944, $p = .068$, $_p^2 = .074$], cumulative score x draw value interaction [$F(1,35) = 0.004$, MSE = 65721, $p = .950$, $_p^2 < .001$], draw value x outcome interaction [$F(2,70) = 1.105$, MSE = 17437, $p = .337$, $_p^2 = .031$], and, the three-way interaction [$F(2,70) = 1.052$, MSE = 25991, $p = .355$, $_p^2 = .022$].

*Shift* proportions were compared across draw value, cumulative score and outcomes in a three-way repeated-measures ANOVA (see Fig 1). *Shift* proportions were significantly modulated by a main effect of draw value [$F(1,35) = 5.165$, MSE = 0.042, $p = .029$, $_p^2 = .129$], a main effect of outcome [$F(1,35) = 8.500$, MSE = 0.069, $p < .001$, $_p^2 = .195$], and, an interaction between draw value x outcome [$F(2,70) = 5.080$, MSE = 0.017, $p = .009$, $_p^2 = .126$]. Here, *win-shift* did not differ as a function of draw value (-1 = 61.99%; +1 = 58.38%), nor did *lose-shift* (-1 = 72.74%; +1 = 72.58%). However, *draw-shift* behaviour was significantly larger when

draws were assigned the value of -1 relative to +1 (73.73% and 63.98%, respectively; Tukey's HSD, p < .05). According to one-sampled t-tests, *draw-shift* behaviour when draws were assigned the value of +1 did not significantly differ from the expected value of 66.6% (63.98%; t [35] = -0.892, *p* = .378) but did show a significant bias in favour of shift behaviour when draws were assigned the value of -1 (73.73%; t[35] = 3.105, *p* = .003). Behaviour following wins showed a significant bias in favour of *stay* (60.18% *shift*; t[35] = -2.214, *p* = .033), whereas behaviour following losses showed a significant bias in favour of *shift* (72.66% *shift*; t[35] = 2.269, *p* = .030).

All other main effects and interactions of the ANOVA were non-significant: cumulative score main effect [$F_{(1,35)} = 0.052$, MSE = 0.048, $p$ = .821, $_p^2$ = .001], draw value x cumulative score interaction [$F_{(1,35)} = 0.097$, MSE = 0.021, $p$ = .758, $_p^2$ = .003], draw value x outcome interaction [$F_{(2,70)} = 0.201$, MSE = 0.017, $p$ = .819, $_p^2$ = .005], and, the three-way interaction [$F_{(2,70)} = 0.333$, MSE = 0.008, $p$ = .718, $_p^2$ = .009].

Experiment 2 produced a significant effect of *draw* value, wherein the degree of shift behaviour was increased when draw trials were assigned the same valence and value as lose trials. This was confirmed by further analysis of the degree of *draw-shift* as a function of draw value (+1, -1) and Experiment (1, 2 [score present conditions only]) via a mixed, two-way ANOVA. The main effect of draw value [$F_{(1,74)} = 9.353$, MSE = .018, $p$ = .003, $_p^2$ = .112] in the absence of a main effect of Experiment [$F_{(1,74)} = 0.558$, MSE = .049, $p$ = .457, $_p^2$ = .007], or interaction with Experiment [$F_{(1,74)} = 0.778$, MSE = .017, p = .381, $_p^2$ = .010] confirms that the proportion of *draw-shift* behaviour increased when draws were assigned the value of -1 relative to +1 (73.25% vs. 66.69%, respectively). Once again, *draw-shift* behaviour exceeded that predicted by MS when draws were assigned a negative value (*t*[75] = 3.630, *p* < .001) but was equivalent to MS behaviour when draws were assigned a positive value (*t*[75] = 0.067, *p* = .947), on the basis of one-sampled t-tests.

### 3.3 Discussion

Across Experiments 1 and 2 we conclude that there is a minor impact of adding 1 to a neutral draw value of 0, relative to a major impact of subtracting 1. This observation is consistent with the view that losses carry twice the subjective magnitude of their objective value [6, 30, 31]. However, if we carry the principle that the negative-going modulation of value should have a greater impact than the positive-going modulation of value *irrespective of the outcome in question*, then increasing the cost of loss from -1 to -2 should produce a stronger effect on *shift* behaviour than increasing the benefit of win from +1 to +2 produces on *stay* behaviour. Therefore, Experiment 3 applied the same value manipulations to win and loss value, once again combined with the presence or absence of cumulative score. In these regards, this final experiment serves as a test that ambiguous outcomes (*draws*) are more sensitive to value manipulations than explicit outcomes (*wins*, *losses*).

## 4. Experiment 3

### 4.1 Method

36 participants were analyzed for Experiment 3. For the 35 individuals who provided demographic information, 24 were women, 30 were right-handed, and the mean age was 19.83 (*sd* = 2.85). All parameters in Experiment 3 were identical to Experiment 2, apart from the value assignments of outcomes. In one condition, wins, draws and losses were assigned +2, 0, -1, respectively, while in a second condition, they were assigned +1, 0, -2, respectively (*win-heavy*, and, *loss-heavy*; after Forder & Dyson, 2016).

**Table 3. Distribution of item, outcome, outcome-response contingency and reaction time as a function of the value of win (+2) and lose (-2) trials and presence (S) or absence (NS) of cumulative score in Experiment 3.** Standard error in parenthesis.

| | | Item | | | Outcome | | |
|---|---|---|---|---|---|---|---|
| | | *Rock* | *Paper* | *Scissors* | *Win* | *Lose* | *Draw* |
| W+2 | S | .378 (.023) | .318 (.016) | .303 (.015) | .335 (.009) | .335 (.010) | .330 (.007) |
| W+2 | NS | .356 (.015) | .353 (.013) | .292 (.015) | .332 (.006) | .324 (.008) | .344 (.007) |
| L-2 | S | .370 (.022) | .334 (.014) | .296 (.014) | .332 (.010) | .342 (.008) | .325 (.008) |
| L-2 | NS | .347 (.014) | .352 (.011) | .301 (.013) | .327 (.008) | .337 (.007) | .335 (.008) |
| | | Outcome-Response Contingency | | | Reaction Time (ms) | | |
| | | *Win-Stay* | *Lose-Shift* | *Draw-Shift* | *Win* | *Lose* | *Draw* |
| W+2 | S | .412 (.048) | .763 (.040) | .676 (.036) | 651 (115) | 567 (129) | 491 (73) |
| W+2 | NS | .363 (.040) | .753 (.031) | .694 (.033) | 489 (74) | 399 (45) | 499 (88) |
| L-2 | S | .427 (.045) | .783 (.031) | .667 (.033) | 665 (98) | 472 (66) | 470 (52) |
| L-2 | NS | .418 (.043) | .786 (.025) | .668 (.036) | 508 (78) | 410 (46) | 406 (48) |

## 4.2 Results

As in all previous experiments, the occurrence of *wins* (33.16%), *losses* (33.46%) and *draws* (33.38%) was equivalent [$F(2,70) = 0.099$, MSE $< .001$, $p = .906$, $_p^2 = .002$; see Table 3]. Median RTs from trial *n* were submitted to a three-way repeated-measures ANOVA featuring value (*win-heavy*, *lose-heavy*) and cumulative score (*present*, *absent*) and outcome at trial *n-1* (*win*, *lose*, *draw*). The two-way interaction between cumulative score x outcome found in Experiment 2 replicated in Experiment 3: [$F(2,70) = 3.340$, MSE $= 48207$, $p = .041$, $_p^2 = .087$] in addition to a main effect of outcome: [$F(1,35) = 7.321$, MSE $= 85044$, $p = .001$, $_p^2 = .173$].

As in Experiment 2, *post-loss speeding* (519 ms) and *post-draw speeding* (481 ms) relative to RTs following wins (658 ms) were only observed in the condition where the cumulative score was present (Tukey's HSD, *p*s $< .004$). Reaction times when the cumulative score was present following wins were also slower than all outcomes when the cumulative score was absent (498, 405, 453 ms for wins, losses and draws, respectively). *Post-loss speeding* was also significant in the cumulative score absent condition (Tukey's HSD, $p = .030$) and all other comparisons were non-significant.

All other ANOVA main effects and interactions were not significant: cumulative score main effect [$F(1,35) = 3.361$, MSE $= 325091$, $p = .075$, $_p^2 = .088$], value main effect [$F(1,35) = 0.553$, MSE $= 149089$, $p = .462$, $_p^2 = .016$], cumulative score x value interaction [$F(1,35) = 0.010$, MSE $= 434503$, $p = .919$, $_p^2 < .001$], value x outcome interaction [$F(2,70) = 1.357$, MSE $= 39730$, $p = .264$, $_p^2 = .037$], and, the three-way interaction [$F(2,70) = 2.298$, MSE $= 31233$, $p = .108$, $_p^2 = .062$].

*Shift* proportions were analysed across value (*win-heavy*, *lose-heavy*),cumulative score (*present*, *absent*) and outcome (*win*, *lose*, *draw*) in a three-way repeated-measures ANOVA (see Fig 1). A main effect of outcome was noted [$F(2,70) = 14.632$, MSE $= 0.077$, $p < .001$, $_p^2 = .295$], along with an interaction between value x outcome [$F(2,70) = 3.597$, MSE $= 0.010$, $p = .033$, $_p^2 = .093$]. None of the pairwise comparisons between value were significant (*win*: *win-heavy* = 57.76% vs *lose-heavy* = 61.27%; *lose*: *win-heavy* 78.44% vs. *lose-heavy* = 75.80%; *draw*: *win-heavy* = 66.73% vs. *lose-heavy* = 68.52%; all Tukey's HSD, p $> .05$). Aggregate behaviour following *win* was not significantly different from the expected value of 66.6% (59.51%; $t[35] = -1.909$, $p = .065$), aggregated behaviour following *lose* was significantly biased towards *shift* (77.12%; $t[35] = 4.174$, $p < .001$), and aggregated behaviour following *draw* was not significantly different from the expected value of 66.6% (67.62%; $t[35] = 0.363$, $p = .719$).

### 4.3 Discussion

The data from Experiment 3 show that wins and losses do not respond to value manipulations in the same way as draws (Experiments 1 and 2). Contrary to expectations, increasing the cost of loss from -1 to -2 did not produce a stronger effect on *shift* behaviour than increasing the benefit of win from +1 to +2 in order to change *stay* behaviour (contra [22]). As such, ambiguous outcomes (*draws)* appear more amenable to value manipulations than explicit outcomes (*wins*, *losses*). Moreover, this cannot be due to the average absolute score achieved in the different value conditions. The assignment of values as win +1, draw +1, lose -1 in Experiment 2 (mean final score = 29.67, *standard error* = 1.03) was functionally equivalent to the assignment of values as win +2, draw 0, lose -1 in the *win-heavy* condition of Experiment 3 (mean final score = 30.35, *standard error* = 1.37), just as the assignment of values as win +1, draw -1, lose -1 in Experiment 2 (mean final score = -30.86, *standard error* = 1.00) was functionally equivalent to the assignment of values as win +1, draw 0, lose -2 in the *lose-heavy* condition of Experiment 3 (mean final score = -31.44, *standard error* = 1.37). This was supported by the results of a two-way, repeated measures ANOVA with final score as the dependent variable. This yielded a main effect of value [F(1,70) = 2163.64, MSE = 62.2, $p < .001$, $_p^2$ = .969] in the absence of main effect of experiment [F(1,70) = 0.002, MSE = 47.4, $p = .966$, $_p^2 < .001$], and in the absence of an interaction [F(1,70) = 0.231, MSE = 62.2, $p = .632$, $_p^2$ = .003].

Although the presence / absence of a cumulative score does not play a role in the *quality* of decision-making, a consistent picture emerges from Experiments 2–3 in that the presentation of an increasingly positive-going / negative-going score plays a role in the *speed* of decision-making- namely, accentuating *speeding following 'negative' outcomes*. This reinforces the idea that individual moments of feedback are compared against a backdrop of historic performance (c.f., [39]), reminding us that reactions to outcome are idiosyncratic, and influenced by the salience of both internal and external signals.

It remains possible that the comparison between Experiments 2 and 3 is not exact, as the subjective change from 0 to +1 (or -1) in the case of *draws* is not as dramatic as the subjective change from -1 to -2 in the case of *losses*, or, from +1 to +2 in the case of *wins* (cf, Prospect Theory; [30]). Indeed, more radical pay-off matrices regarding the value of wins and losses can be proposed (eg [40]). However, such refinements will reintroduce the confound of different average absolute scores across different conditions. Thus, the aim to equate the *subjective* rather than objective value of outcomes also has a number of hidden assumptions that will require consideration.

## 5. General discussion

In the absence of explicit reinforcement or punishment, we must still evaluate the relative success or failure of our actions and prepare for future behaviour accordingly. The current data speak clearly to outcome instances which are- on the surface- neither gains nor losses.

Previous data have strongly argued for the *draw* experience to elicit negative rather than positive responses. For example, models of human performance are better when draws are directly punished in the same way as losses (that is, assigned the value of -1), rather than being neither punished nor rewarded (assigned the value of 0; [38]). Similar profiles for draws and losses are shown in terms of a speeded bias towards shift behaviour, further implying their subjective interpretation as negative rather than positive (e.g., [22, 41]). While our data also support the contention that responses following draws will be initiated faster than those following explicit wins, and, that draws produce a future response *switch* (rather than *stay*) bias, the experience of draws is not wholly negative. This is borne out in the logic that fast rather than slow reaction times following loss represent a self-imposed limitation on future decision-

making time that makes automatic, sub-optimal performance more likely ([10, 18, 31]). However, the *post-draw speeding* observed in the current experimental series did not produce the same sub-optimal performance as that observed for *post-loss speeding*. The average proportion of *draw-shift* behaviour was equivalent to that predicted by mixed-strategy performance wherein all three possible response associations between consecutive trials were equal (*draw-stay* ≈ 33.3%; *draw-shift* ≈ 66.6%). Such performance *guarantees* loss minimization and matches the approximation of mixed-strategy performance often observed following wins.

Moreover, draws allowed for a greater degree of behavioural flexibility relative to outcomes that were clearly marked as negative. Our data show that changing the value of draws also leads to changes in behaviour- specifically, assigning *draws* the value of explicit *wins* (+1) enabled an approximation of optimal behaviour (see above). In contrast, assigning *draws* the value of explicit *losses* (-1) lead to an increase in *shift* bias thereby placing the participant in a potentially precarious and exploitable competitive position (see Fig 1). These comparisons strongly suggest another way in which draws can behave like wins rather than losses. The suggestion that the *draw* outcome as chameleon-like in nature, however, must be tempered by additional observations. Based on a reviewer's suggestion, we examined the degree of *shift* behaviour following wins (+1; 61.99%) in Experiments 1 and 2 against the concomitant behaviour generated by draws (+1; 66.62%) within those same conditions. We noted a significant reduction in *win-shift* behavior relative to *draw-shift* behaviour, when both categories were assigned a constant value of +1 $t[75] = -3.051$, $p = .003$. Similarly, we examined the degree of *shift* behaviour following losses (-1; 74.10%) in Experiments 1 and 2 against the concomitant behaviour generated by draws (-1; 73.69%), we noted a non-significant difference in lose-shift behavior relative to draw-shift behaviour, when both categories were assigned a constant value of -1: $t[76] = 0.292$, $p = .771$. Our reading of these results are that *draw* outcomes may more readily co-opt the behavioural signatures of explicitly negative, relative to explicitly positive, outcomes when the same values are assigned. If a +1 *draw* does not have the exact same properties as a +1 *win*, then value in and of itself cannot completely predict the behavioural consequences of outcomes.

Thus, the study of neutral outcomes will continue to represent an important addition to decision-making for two reasons. First, our current data imply a cognitive flexibility following draw trials that is not triggered by more clearly valenced wins and losses. This provides a clear route to studying the subjective aspects of outcome response, and how reaction to neutral trials may be shaped by preceding trial history. For example, we have recently shown that relative to *draw-draw* trials, the trial outcome sequence of *win-draw* causes an increase in shift behaviour whereas the sequences of *lose-draw* causes a decrease in shift behaviour- at least at a group level [11]. The concomitant increase and decrease in shift behaviour appears to us as an objective manifestation of the subjective nature of signed prediction error theory (eg, [42, 43]). In other words, draw trials preceded by a win are interpreted as *worse-than-expected* (making the draw appear more negative) whereas draw trials preceded by a loss are interpreted as *better-than-expected* (making the draw appear more positive). Second, introducing a third outcome (*draw*) requires a concomitant increase in the number of responses within the decision-making space. This is important as there are growing concerns as to the degree to which the modal use of binary decision-making paradigms severely limits our understanding of more naturalistic, non-binary decision spaces [44, 45].

In sum, mitigating the adverse emotional and cognitive consequences of negative outcomes remains an important goal in the context of education, gambling and economics. However, our data show that moving away from binary conceptualizations of outcome is critical to understanding the full palette of subjective responses elicited by decision-making. Specifically, our consideration of *draws* highlights the subjective aspects of decision-making, and the ways

in which supposedly neutral outcomes take on the hues of more clearly valenced results. The processing cascade generated by neither being explicitly reinforced or punished produces a complex behavioural profile containing elements found in both explicitly positive and explicitly negative results. The reaction to draws appears more flexible than those produced by wins and losses, and generates a response signature that is simultaneously positive and negative, but apparently never 'neutral'.

## Supporting information

**S1 Data.**
(XLSX)

## Author Contributions

**Conceptualization:** Benjamin James Dyson.

**Formal analysis:** Benjamin James Dyson.

**Funding acquisition:** Benjamin James Dyson.

**Investigation:** Rimsa Dahal, Kelsey MacLellan, Danielle Vavrek.

**Methodology:** Benjamin James Dyson.

**Project administration:** Rimsa Dahal, Kelsey MacLellan, Danielle Vavrek.

**Writing – original draft:** Benjamin James Dyson.

**Writing – review & editing:** Rimsa Dahal, Kelsey MacLellan, Danielle Vavrek.

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
