## [Decision Letter · Decision Letter 0]

26 Apr 2022

PONE-D-22-06419Assessing the behavioural profiles following neutral, positive and negative feedbackPLOS ONE

Dear Dr. Dyson,

Thank you for submitting your manuscript to PLOS ONE. After careful consideration, we feel that it has merit but does not fully meet PLOS ONE’s publication criteria as it currently stands. Therefore, we invite you to submit a revised version of the manuscript that addresses the points raised during the review process.

We look forward to receiving your revised manuscript.

Kind regards,

Alberto Antonioni, PhD

Academic Editor

PLOS ONE

Journal Requirements:

Additional Editor Comments:

I recommend a major revision following both reviewers' suggestions to improve the manuscript before being considered it for publication. In particular, all the comments by Reviewer 2 should be addressed.

Reviewers' comments:

Reviewer's Responses to Questions

**Comments to the Author**

1. Is the manuscript technically sound, and do the data support the conclusions?

Reviewer #1: Yes

Reviewer #2: Partly

2. Has the statistical analysis been performed appropriately and rigorously? 

Reviewer #1: Yes

Reviewer #2: Yes

3. Have the authors made all data underlying the findings in their manuscript fully available?

Reviewer #1: Yes

Reviewer #2: Yes

4. Is the manuscript presented in an intelligible fashion and written in standard English?

Reviewer #1: Yes

Reviewer #2: Yes

5. Review Comments to the Author

Reviewer #1: The current paper presents a neat set of experiments that adds nicely to the literature on post-error/post-loss responses by adding a third possible outcome such as a draw. This draw somewhat resembles near-misses in the gambling literature, however not entirely and therefore is a good addition in order to investigate decision making in response to different outcomes.

I just have a couple of points that the authors might want to take into consideration:

On the theoretical level, I am not entirely sure whether the frequency argument made by the orienting account is the only explanation for speeding vs. slowing. In addition to the fact, that Eben et al. (2020) observed speeding after losses with an equal amount of wins and losses (which seems so do you), recent research (e.g. Damaso et al. 2020) suggests that speeding vs. slowing might be determined by the controllability of the outcome (which in turn allows the idea that speeding is not necessarily maladaptive ...).

Personally, I would find a sample size rationale helpful.

I am curious why the authors chose to determine whether rock, paper or scissors were chosen instead of predetermining the outcome to be win, draw or loss? We usually followed the latter approach to be sure we indeed have an equal amount of outcomes, so I am happy to hear other options.

In the spirit of openness I always encourage authors not only to share their data but also the analyses scripts and the material which increases the reproducibility of your experiments and analyses.

Reviewer #2: The current paper reports three experiments that examined the behavioral profiles following wins, losses and draws. The effects of draws on subsequent responses are interesting, and as noted by the authors, certainly largely neglected in the literature so far. From this perspective, the current paper makes a useful contribution. However, the way the data is analyzed and the results are presented seem to hinder the understanding of the findings. Furthermore, I am not totally convinced that based on the results, the authors can claim that draws differ from both wins and losses because they show different sensitivity to value manipulation (see major comment 8). Hope my comments will be helpful to the authors in preparing the manuscript for publication.

Major comments:

1. First of all, I would like to commend the authors for making the aggregate data publicly available on the Open Science Framework. Aggregate data will allow other researchers to check the accuracy of the reported results, and is thus probably one of the most important products of a research project. That being said, other products generated during the process of research are also important, and I would encourage the authors to publicly share these products as much as possible. For instance, these may include the experimental materials, the Presentation code for the experiment, raw data files, analysis code used to analyze data etc. By making these materials open, other researchers will be able to repeat the analysis, re-analyze the raw data in new ways, or run replications and extensions in the future. I think this will be essential for a cumulative and robust science.

2. The introduction set the stage up very well to highlight the importance of examining 'draw' outcomes, a type of outcome that has largely been neglected so far in the literature. The following paragraphs on reaction times, FRN and shift/stay behavior also did an excellent job of reviewing relevant previous work. However, what seem to be missing are clear statements of the research questions that the current paper aims to address. After learning that both RT and FRN show similar patterns after a draw and a loss, it seems clear that draws are more like unambiguously negative loss outcomes. If that is the case, what are the remaining questions or gaps that the current paper aims to address? A paragraph at the end of the introduction on what remains unknown, and what this paper will examine, would be helpful.

3. The sample sizes of the experiments seem relatively low (N = 40, 36 and 36), especially given that the designs involve multiple factors and interaction effects are of interest (see e.g., Brysbaert, M. (2019). How Many Participants Do We Have to Include in Properly Powered Experiments? A Tutorial of Power Analysis with Reference Tables. Journal of Cognition, 2(1), 1–38. https://doi.org/10.5334/joc.72). Of course, whether a certain sample size provides sufficient statistical power or not depends on the expected effect size. I would therefore like the authors to provide more information on how the sample sizes were determined, and the achieved statistical power.

4. Figure 1 (and the corresponding analyses on stay/shift behavior) is difficult to follow, partly because it shows the proportion of stay for the win condition, and the proportion of shift for the lose and draw conditions. While stay and shift are complementary (they add up to 100%), they are not directly comparable. That the values in the win-stay condition are lower than the values in the lose-shift and draw-shift conditions in Figure 1 is thus not very informative. I would suggest showing the proportion of shift (or the proportion of stay) for all three outcomes. The authors may then add a horizontal line at 66.6%, showing the expected shift proportion of a player who chooses randomly. Values higher than 66.6% indicate a tendency to shift, while values lower than 66.6% indicate a tendency to stay.

5. The analyses on stay/shift behavior can be organized accordingly. For Experiment 1 for example, this would mean to start with a 3 (previous outcome, win, loss vs. draw) by 3 (draw value, 0, 1 vs. -1) ANOVA on the proportion of shift, followed up by separate ANOVAs or t tests if necessary. The proportions of shift can then be compared against the baseline of 66.6%, to establish (1) whether they differ from the baseline, and if yes, (2) in which direction (stay vs. shift). Note that in the approach adopted by the authors, they needed to look at the shift behavior after wins (in the very last analysis) anyways. I think consistently using the proportion of shift as the dependent variable may provide a better structure for the analyses and the results.

6. Statistical results are selectively reported, in the sense that for ANOVAs, only the details for significant effects but not non-significant effects are presented. This is OK for the sake of brevity and simplicity, but I think it is still important to show the detailed results for all effects from all analyses, regardless of statistical significance. For instance, the authors may report the full results in the Supplemental Materials and refer readers to it in the main text should they be interested. Related, some of the conclusions are based on non-significant results. However, absence of evidence is not evidence of absence. Non-significant results can also mean that the evidence is inconclusive (given the relatively small sample sizes), rather than supporting the null hypothesis. To obtain evidence for the absence of an effect, the authors will need to adopt other statistical approaches, such as Bayesian analysis.

7. In the discussion of Experiment 2, the authors noted that "losses carry twice the subjective magnitude of their objective value". My understanding of this statement is that in the value function in the Prospect theory, the decrease in subjective value from 0 to -1 is about twice as large as the increase in subjective value from 0 to 1. Thus, losses carry twice the subjective magnitude, in comparison to wins. I'm not sure if this would necessarily imply that the difference between -2 and -1 in the loss domain would still be twice as large as the difference between 2 and 1 in the win domain. After all, the value function has an S shape, and as the magnitude of wins and losses increases, any additional win or loss will have an increasingly smaller influence on the subjective value.

8. Related, one of the conclusions of the paper is that "wins and losses do not respond to value manipulations in the same way as draws". However, changing the value from 1 to 2 or from -1 to -2 is probably different from changing the value from -1 to 1. So I am not sure if the authors can ascribe the difference to wins and losses being different from draws. To make such a claim, a better test may be to give a value of -1 or -2 to a draw , and compare it to e.g. loss when the value for a loss is also varied from -1 to -2. Some of the data may already provide some initial insights into this issue. For instance, what results would the authors get from Experiments 1 and 2 if they would compare shift/stay between (a) draw (-1) and loss (-1) and (b) draw (+1) and win (+1)? When the values are matched, are there still any differences between the different types of outcomes?

Minor comments:

1. Page 4: While the rest of the paper mainly used the term "post-loss speeding" (which is also the term used in Verbruggen et al. and Eben et al.,), the term "post-error slowing/speeding" was used here. Losses and errors may both be seen as failures or sub-optimal outcomes, but they are often used in different contexts: losses often denote losing a reward, while errors indicate incorrect responses in mostly cognitive psychology tasks (and may not be explicitly rewarded or punished). I think a short discussion/clarification on how the two may be related would be useful, instead of directly treating them as the same.

2. Page 5: Related, while Notebaert et al. indeed showed that the frequency of errors influences post-error slowing or speeding, Eben et al. (Experiment 3) has observed post-loss speeding while losses, wins and non-gamble outcomes occur equally often. Outcome frequency alone thus does not seem sufficient to explain post-loss speeding.

3. Page 5: "A second metric to ascertain the subjective interpretation of draws is the flexibility of responding following outcomes.". The paragraph then went on to discuss previous findings on feedback-related negativity, rather than response flexibility.

4. Page 6: I feel the paragraph on behavior flexibility would be easier to follow if the authors would first start with the phenomenon (loss-shift is not sensitive to objective values while win-shift appears to be sensitive) and then explain the underlying mechanism (that loss subjectively looms larger than win).

5. Page 6: "a mixed-strategy-style approach guaranteeing the absence of exploitation". Please explain what this strategy is when it first occurs. Does it mean choosing randomly?

6. Page 7: The introduction of Experiment 1 may be substantially shortened, by saying that since outcome frequencies influence reaction times (Notebaert et al.) and potentially the subjective interpretation of draws (e.g., when wins are frequent, draws may be more likely to be interpreted as failing to win), the authors used an opponent that chose randomly to ensure equal probabilities of win, loss and draw.

7. Page 9: One of the predictions for Experiment 1 is that "the degree to which participants stray from optimal performance should be greater for draw-shift than win-stay". My interpretation of this prediction is that the deviation from 66.6% for draw-shift would be larger than the deviation from 33.3% for win-stay. Or, if the authors adopt the suggestion of consistently using the proportion of shift (see above), this would mean the absolute difference between draw-shift and 66.6% would be larger than the absolute difference between win-shift and 66.6%. This does not seem to be tested. Or did I misunderstand this prediction?

8. An inconsistency in the description of the method. Page 9: "white-gloved (left; opponent) and blue-gloved (right; participant)". Page 10: "the opponent (blue glove on the left) and the participant (white glove on the right)".

9. Did participants get a reward based on their performance in the task? In other words, was the task incentivized?

10. In the instructions given to the participants, were there any indications as to whether the opponent could potentially be exploited? For instance, were participants explicitly told that they were playing against the computer, which would select the options randomly?

11. Which statistical software (and version) was used to analyze the data?

12. Page 14: "we note changing the value of a draw to -1 did not significant impact". Significant should be significantly?

13. Figure 1 currently only plots the shift/stay proportions. Since reaction times are of interest as well, plotting RT would be useful. So Figure 1 would contain 6 subplots, with one column for RTs and one column for shift/stay.

14. Page 19: The first paragraph of the Discussion seems better suited for the Results section.

15. Page 23: What is 'sterr'?

16. Page 23: What is the dependent variable in the two-way, repeated-measures ANOVA?

17. When participants shift, there are two different options that they can shift to. I wonder if the authors also looked at different types of shift to see if draw and loss differ.

18. Page 25: "a win followed by a draw causes an increase in shift behaviour whereas a loss followed by a draw causes shift behaviour to decrease". This sounds like a win leads to an increase in shift behavior and a loss leads to a decrease, but I think the authors meant that the draws have such an effect.

6. PLOS authors have the option to publish the peer review history of their article (what does this mean?). If published, this will include your full peer review and any attached files.

Reviewer #1: **Yes: **Charlotte Eben

Reviewer #2: **Yes: **Zhang Chen

---

## [Author Response · Author response to Decision Letter 0]

17 May 2022

16 / 05 / 2022

Thank you for allowing us to revise and resubmit our manuscript “Assessing the behavioural profiles following neutral, positive and negative feedback” (PONE-D-22-06419). In accordance with your guidance, we have revised the manuscript, and, provided detailed point-by-point coverage, of both Reviewers’ comments. 

If you require anything further, please do not hesitate to contact me directly. 

With best wishes, 

Dr Ben Dyson.

Reviewer #1: The current paper presents a neat set of experiments that adds nicely to the literature on post-error/post-loss responses by adding a third possible outcome such as a draw. This draw somewhat resembles near-misses in the gambling literature, however not entirely and therefore is a good addition in order to investigate decision making in response to different outcomes.

I just have a couple of points that the authors might want to take into consideration:

On the theoretical level, I am not entirely sure whether the frequency argument made by the orienting account is the only explanation for speeding vs. slowing. In addition to the fact, that Eben et al. (2020) observed speeding after losses with an equal amount of wins and losses (which seems so do you), recent research (e.g. Damaso et al. 2020) suggests that speeding vs. slowing might be determined by the controllability of the outcome (which in turn allows the idea that speeding is not necessarily maladaptive ...).

We agree that the frequency argument is not the only way to frame the speeding / slowing account. Damaso et al (2020) cites our previous work (Dyson, Sundvall, Forder & Douglas, 2018), where we also claim that impulsive behavior following failure is more likely when an individual cannot control the outcome of their environment either as a result of interacting with an unexploitable opponent, or, interacting with an exploitable opponent whom they fail to exploit. Furthermore, we are starting to see some links at the individual level where the degree of post-error slowing relates to the degree to which the participant can successfully exploit their opponent (Dyson, 2021). 

Similar to the comments made by Reviewer #2 we now make a statement that frequency of outcome cannot provide a complete account of post-loss slowing since post-loss slowing is intact when positive and negative outcomes have equal frequency.

Personally, I would find a sample size rationale helpful.

Similar to the suggestion of Reviewer 2, we now provide a sample size rationale in the manuscript. 

I am curious why the authors chose to determine whether rock, paper or scissors were chosen instead of predetermining the outcome to be win, draw or loss? We usually followed the latter approach to be sure we indeed have an equal amount of outcomes, so I am happy to hear other options.

We have previously used both variable win-rate controlling for the distribution of items (current experiments) and fixed win-rate paradigms controlling for the distribution of outcomes (eg, Dyson, Musgrave, Rowe & Sandhur, 2020). We see two potential disadvantages with fixed win-rate paradigms. First, if participants are completing a series of conditions (in our manuscript, either 3 or 4), there may be symmetries in the final score (eg., always 0, -30 or +30) that might change the way participants approach the study. In other words, fixed win rates can increase the belief that their computerized opponent is somehow ‘cheating’. Second, in cases where the experience of dominance is simulated by increasing win rates to the same degree, the mental model participants develop for their ‘successful strategy’ is idiosyncratic, or, nonexistent. The separation between success and the mechanism(s) for success might lead participants towards a belief that the task was one of luck rather than skill. 

However, there are also disadvantages with the variable win-rate approach. For example, distributing items equally and randomly across a series of trials raises the possibility of accidental exploitation if a high degree of item bias is exhibited in the earlier stages of the block. For example, opponent overplay of rock in the first half of the block requires the underplay of rock in the second half of the block. However, empirical analyses do not support the idea that participants used the degree of preliminary item bias to increase their win rate in ‘unexploitable’ conditions (see Dyson et al., 2018, footnote 1). 

In the long-run, both variable and fixed win rates have their uses as long as we remain aware of the potential pitfalls of both. 

In the spirit of openness I always encourage authors not only to share their data but also the analyses scripts and the material which increases the reproducibility of your experiments and analyses.

Echoing the comments of Reviewer 2, we have now added experimental scripts and raw data to our pre-print and summary data on https://psyarxiv.com/cqeg7/

Reviewer #2: The current paper reports three experiments that examined the behavioral profiles following wins, losses and draws. The effects of draws on subsequent responses are interesting, and as noted by the authors, certainly largely neglected in the literature so far. From this perspective, the current paper makes a useful contribution. However, the way the data is analyzed and the results are presented seem to hinder the understanding of the findings. Furthermore, I am not totally convinced that based on the results, the authors can claim that draws differ from both wins and losses because they show different sensitivity to value manipulation (see major comment 8). Hope my comments will be helpful to the authors in preparing the manuscript for publication.

Major comments:

1. First of all, I would like to commend the authors for making the aggregate data publicly available on the Open Science Framework. Aggregate data will allow other researchers to check the accuracy of the reported results, and is thus probably one of the most important products of a research project. That being said, other products generated during the process of research are also important, and I would encourage the authors to publicly share these products as much as possible. For instance, these may include the experimental materials, the Presentation code for the experiment, raw data files, analysis code used to analyze data etc. By making these materials open, other researchers will be able to repeat the analysis, re-analyze the raw data in new ways, or run replications and extensions in the future. I think this will be essential for a cumulative and robust science.

We thank the Reviewer for the reminder to improve our commitment to Open Science. Similar to the comments of Reviewer #1, we have now added experimental scripts and raw data to our pre-print and summary data on https://psyarxiv.com/cqeg7/

2. The introduction set the stage up very well to highlight the importance of examining 'draw' outcomes, a type of outcome that has largely been neglected so far in the literature. The following paragraphs on reaction times, FRN and shift/stay behavior also did an excellent job of reviewing relevant previous work. However, what seem to be missing are clear statements of the research questions that the current paper aims to address. After learning that both RT and FRN show similar patterns after a draw and a loss, it seems clear that draws are more like unambiguously negative loss outcomes. If that is the case, what are the remaining questions or gaps that the current paper aims to address? A paragraph at the end of the introduction on what remains unknown, and what this paper will examine, would be helpful.

In terms of providing evidence that draws may generate a unique behavioural fingerprint relative to losses, we now identify a seemingly reliable observation from our previous work in the Introduction. Specifically, the degree of shift behaviour is reliably and significantly smaller following draws relative to losses (cf, Dyson, 2021, Experiment 1, no credit condition; Dyson, Stewart, Meneghetti & Forder, 2020, Experiment 1; Forder & Dyson, 2016, baseline condition; Dyson, Wilbiks, Sandhu, Papanicolaou & Lintag, 2016). These previous observations prohibit the wholesale acceptance of draws as functionally equivalent to losses, justify the further study of draw trials, and provide a rationale for our sample sizes (see below). 

3. The sample sizes of the experiments seem relatively low (N = 40, 36 and 36), especially given that the designs involve multiple factors and interaction effects are of interest (see e.g., Brysbaert, M. (2019). How Many Participants Do We Have to Include in Properly Powered Experiments? A Tutorial of Power Analysis with Reference Tables. Journal of Cognition, 2(1), 1–38. https://doi.org/10.5334/joc.72). Of course, whether a certain sample size provides sufficient statistical power or not depends on the expected effect size. I would therefore like the authors to provide more information on how the sample sizes were determined, and the achieved statistical power.

Sample sizes were derived with 80% estimates of power (cf, Brysbaert, 2019) using G*Power (Faul, Erdfelder, Lang & Buchner, 2007) from two recent studies described above, in which reduction in shift behaviour following draw (0) relative to loss (-1) trials were observed: Dyson (2021): dz = 0.6894, one-tailed, yielding n = 15, and, Dyson, Stewart, Meneghetti & Forder (2020): dz = 0.4217, one-tailed, yielding n = 37. 

4. Figure 1 (and the corresponding analyses on stay/shift behavior) is difficult to follow, partly because it shows the proportion of stay for the win condition, and the proportion of shift for the lose and draw conditions. While stay and shift are complementary (they add up to 100%), they are not directly comparable. That the values in the win-stay condition are lower than the values in the lose-shift and draw-shift conditions in Figure 1 is thus not very informative. I would suggest showing the proportion of shift (or the proportion of stay) for all three outcomes. The authors may then add a horizontal line at 66.6%, showing the expected shift proportion of a player who chooses randomly. Values higher than 66.6% indicate a tendency to shift, while values lower than 66.6% indicate a tendency to stay.

In accordance with the Reviewer’s suggestions, Figure 1 has been redrawn for shift proportion throughout, in addition to the plotting of a baseline of 66.6%. 

5. The analyses on stay/shift behavior can be organized accordingly. For Experiment 1 for example, this would mean to start with a 3 (previous outcome, win, loss vs. draw) by 3 (draw value, 0, 1 vs. -1) ANOVA on the proportion of shift, followed up by separate ANOVAs or t tests if necessary. The proportions of shift can then be compared against the baseline of 66.6%, to establish (1) whether they differ from the baseline, and if yes, (2) in which direction (stay vs. shift). Note that in the approach adopted by the authors, they needed to look at the shift behavior after wins (in the very last analysis) anyways. I think consistently using the proportion of shift as the dependent variable may provide a better structure for the analyses and the results.

Full factorial ANOVAs are applied to Experiments 1-3. One-sampled t-tests comparing observed values to the baseline of 66.6% are also applied to assess any direction of bias. 

6. Statistical results are selectively reported, in the sense that for ANOVAs, only the details for significant effects but not non-significant effects are presented. This is OK for the sake of brevity and simplicity, but I think it is still important to show the detailed results for all effects from all analyses, regardless of statistical significance. For instance, the authors may report the full results in the Supplemental Materials and refer readers to it in the main text should they be interested. Related, some of the conclusions are based on non-significant results. However, absence of evidence is not evidence of absence. Non-significant results can also mean that the evidence is inconclusive (given the relatively small sample sizes), rather than supporting the null hypothesis. To obtain evidence for the absence of an effect, the authors will need to adopt other statistical approaches, such as Bayesian analysis.

All statistical terms arising from the full factorial ANOVAs (see 5) are now included in the manuscript. In the context of cross-experimental comparisons (such as those suggested by the Reviewer; see below), we feel that a significant difference between Experiments removes some of the framing concerns related to ‘non-significant results.’ Moreover, it is not the case that specific experiments was so insensitive as to yield no significant results. For example, in Experiment 3, we replicated the relationship between post-loss speeding and the presence (rather than absence) of cumulative score. We also replicated the observations that behaviour following loss was significantly biased towards shift (see also Experiment 2). Rather, it seems to us more appropriate to discuss the manuscript in terms of the strength of specific manipulations that we deployed (also, see below). 

7. In the discussion of Experiment 2, the authors noted that "losses carry twice the subjective magnitude of their objective value". My understanding of this statement is that in the value function in the Prospect theory, the decrease in subjective value from 0 to -1 is about twice as large as the increase in subjective value from 0 to 1. Thus, losses carry twice the subjective magnitude, in comparison to wins. I'm not sure if this would necessarily imply that the difference between -2 and -1 in the loss domain would still be twice as large as the difference between 2 and 1 in the win domain. After all, the value function has an S shape, and as the magnitude of wins and losses increases, any additional win or loss will have an increasingly smaller influence on the subjective value.

We thank the Reviewer for this comment, and have integrated a discussion of this into the revised manuscript. In particular, we discuss the tension between assigning outcome values that maintain long-run equivalent between cumulative scores across Experiments 2 and 3 (and potentially decreasing subjective value), and, assigning outcome values that confound long-run equivalent between cumulative scores across Experiments 2 and 3 (but potentially equate subjective value). We feel the former approach as a valid one as it ensures that the differences we observed cannot be due to the average absolute score achieved in the different value conditions. As we mention in our original version of our manuscript: 

“The assignment of values as win +1, draw +1, lose -1 in Experiment 2 (mean final score = 29.67, standard error = 1.03) was functionally equivalent to the assignment of values as win +2, draw 0, lose -1 in the win-heavy condition of Experiment 3 (mean final score = 30.35, standard error = 1.37), just as the assignment of values as win +1, draw -1, lose -1 in Experiment 2 (mean final score = -30.86, standard error = 1.00) was functionally equivalent to the assignment of values as win +1, draw 0, lose -2 in the lose-heavy condition of Experiment 3 (mean final score = -31.44, standard error = 1.37). This was supported by the results of a two-way, repeated measures ANOVA with final score as the dependent variable. This yielded, observing a main effect of value [F(1,70) = 2163.64, MSE = 62.2, p < .001, ƞp2 = .969] in the absence of main effect of experiment [F(1,70) = 0.002, MSE = 47.4, p = .966, ƞp2 < .001], and in the absence of an interaction [F(1,70) = 0.231, MSE = 62.2, p = .632, ƞp2 = .003].”

8. Related, one of the conclusions of the paper is that "wins and losses do not respond to value manipulations in the same way as draws". However, changing the value from 1 to 2 or from -1 to -2 is probably different from changing the value from -1 to 1. So I am not sure if the authors can ascribe the difference to wins and losses being different from draws. To make such a claim, a better test may be to give a value of -1 or -2 to a draw, and compare it to e.g. loss when the value for a loss is also varied from -1 to -2. Some of the data may already provide some initial insights into this issue. For instance, what results would the authors get from Experiments 1 and 2 if they would compare shift/stay between (a) draw (-1) and loss (-1) and (b) draw (+1) and win (+1)? When the values are matched, are there still any differences between the different types of outcomes?

Once again, we thank the Reviewer for this suggestion. We examined the degree of shift behaviour following wins (+1; 61.99%) in Experiments 1 and 2 against the concomitant behaviour generated by draws (+1; 66.62%) within those same conditions. We noted a significant reduction in win-shift behavior relative to draw-shift behaviour, when both categories were assigned a constant value of +1 t[75] = -3.051, p =.003. Similarly, we examined the degree of shift behaviour following losses (-1; 74.10%) in Experiments 1 and 2 against the concomitant behaviour generated by draws (-1; 73.69%), we noted a non-significant difference in lose-shift behavior relative to draw-shift behaviour, when both categories were assigned a constant value of -1: t[76] = 0.292, p =.771. Our reading of these results are that a) draw outcomes may more readily co-opt the behavioural signatures of explicitly negative relative to explicitly positive outcomes, and, b) value in and of itself cannot completely predict the behavioural consequences of outcomes. 

Minor comments:

1. Page 4: While the rest of the paper mainly used the term "post-loss speeding" (which is also the term used in Verbruggen et al. and Eben et al.,), the term "post-error slowing/speeding" was used here. Losses and errors may both be seen as failures or sub-optimal outcomes, but they are often used in different contexts: losses often denote losing a reward, while errors indicate incorrect responses in mostly cognitive psychology tasks (and may not be explicitly rewarded or punished). I think a short discussion/clarification on how the two may be related would be useful, instead of directly treating them as the same.

To avoid confusion, we have removed explicit references to ‘post-error’ and now focus on ‘post-loss’ speeding. 

2. Page 5: Related, while Notebaert et al. indeed showed that the frequency of errors influences post-error slowing or speeding, Eben et al. (Experiment 3) has observed post-loss speeding while losses, wins and non-gamble outcomes occur equally often. Outcome frequency alone thus does not seem sufficient to explain post-loss speeding.

Similar to the comments made by Reviewer #1, we now make a statement that frequency of outcome cannot provide a complete account of post-loss slowing since post-loss slowing is intact when positive and negative outcomes have equal frequency. 

3. Page 5: "A second metric to ascertain the subjective interpretation of draws is the flexibility of responding following outcomes.". The paragraph then went on to discuss previous findings on feedback-related negativity, rather than response flexibility.

We have now changed this to ‘neural responding’

4. Page 6: I feel the paragraph on behavior flexibility would be easier to follow if the authors would first start with the phenomenon (loss-shift is not sensitive to objective values while win-shift appears to be sensitive) and then explain the underlying mechanism (that loss subjectively looms larger than win).

The paragraph has been rewritten with this new structure. 

5. Page 6: "a mixed-strategy-style approach guaranteeing the absence of exploitation". Please explain what this strategy is when it first occurs. Does it mean choosing randomly?

The programming of the computerized opponent is now described in that section. 

6. Page 7: The introduction of Experiment 1 may be substantially shortened, by saying that since outcome frequencies influence reaction times (Notebaert et al.) and potentially the subjective interpretation of draws (e.g., when wins are frequent, draws may be more likely to be interpreted as failing to win), the authors used an opponent that chose randomly to ensure equal probabilities of win, loss and draw.

The Introduction to Experiment 1 has now been edited with the above revisions in mind. 

7. Page 9: One of the predictions for Experiment 1 is that "the degree to which participants stray from optimal performance should be greater for draw-shift than win-stay". My interpretation of this prediction is that the deviation from 66.6% for draw-shift would be larger than the deviation from 33.3% for win-stay. Or, if the authors adopt the suggestion of consistently using the proportion of shift (see above), this would mean the absolute difference between draw-shift and 66.6% would be larger than the absolute difference between win-shift and 66.6%. This does not seem to be tested. Or did I misunderstand this prediction?

Now that the dependent variable of interest in shift behaviour following wins, losses and draws, the case (as in Experiment 1) where the average win-shift value of 64.35% does not significantly differ from the expected value of 66.6%, but the average lose-shift value of 76.17% was significantly larger than the expected value of 66.6%, is taken as greater deviation from optimal performance following losses.

8. An inconsistency in the description of the method. Page 9: "white-gloved (left; opponent) and blue-gloved (right; participant)". Page 10: "the opponent (blue glove on the left) and the participant (white glove on the right)".

This is now corrected “(blue-gloved (left; opponent) and white-gloved (right; participant)”

9. Did participants get a reward based on their performance in the task? In other words, was the task incentivized?

The task was not incentivized. 

10. In the instructions given to the participants, were there any indications as to whether the opponent could potentially be exploited? For instance, were participants explicitly told that they were playing against the computer, which would select the options randomly?

For each condition, as part of the instructions participants were informed, “Your opponents may play in different ways and use different strategies to try and win.”

11. Which statistical software (and version) was used to analyze the data?

We now list Statistica 13.3 as the software used to analyze the data. 

12. Page 14: "we note changing the value of a draw to -1 did not significant impact". Significant should be significantly?

Changed to ‘significantly’. 

13. Figure 1 currently only plots the shift/stay proportions. Since reaction times are of interest as well, plotting RT would be useful. So Figure 1 would contain 6 subplots, with one column for RTs and one column for shift/stay.

Reaction Time data are now also provided as a separate column in Figure 1. 

14. Page 19: The first paragraph of the Discussion seems better suited for the Results section.

The first paragraph of the Discussion (3.3) has now been included in the Results section (3.2).

15. Page 23: What is 'sterr'?

Standard error. 

16. Page 23: What is the dependent variable in the two-way, repeated-measures ANOVA?

The manuscript has been updated to reflect that final score is the dependent variable. 

17. When participants shift, there are two different options that they can shift to. I wonder if the authors also looked at different types of shift to see if draw and loss differ.

We have previously identified the selection of an item at trial n + 1 that would have beaten the previous item at trial n (e.g., rock followed by paper), or, the selection of an item in trial n + 1 that would have been beaten by the previous item at trial n (e.g., rock followed by scissors). In the first instance, such behavior is described as an ‘ascending’, ‘right-shift’, ‘one-ahead’ or ‘upgrade’ strategy, whereas in the second instance, behavior has been described as a ‘descending’, ‘left-shift’ or ‘downgrade’ strategy (Baek et al., 2013; Stöttinger, Filipowicz, Danckert & Anderson, 2014; Wang & Xu, 2014). In terms of behavioural data, there is weak evidence that losses initiate more downgrade responses and draws initiate more upgrade responses (Dyson, Wilbiks, Sandhu, Papanicolaou & Lintag, 2016). However, a close examination of the framing of such specific ‘strategies’ is problematic in terms of the isomorphism with other- perhaps simpler- responses. The framing problem of strategy is discussed in Dyson (2019). 

18. Page 25: "a win followed by a draw causes an increase in shift behaviour whereas a loss followed by a draw causes shift behaviour to decrease". This sounds like a win leads to an increase in shift behavior and a loss leads to a decrease, but I think the authors meant that the draws have such an effect.

We have now re-written this section to be clearer that the trial outcome sequence of win-draw causes an increase in shift behaviour, whereas the sequence of lose-draw causes a decrease in shift behaviour. 

References

Baek, K.; Kim, Y.-T.; Kim, M.; Choi, Y.; Lee, M.; Lee, K.; Hahn, S. & Jeong, J. (2013). Response randomization of one-and two-person Rock-Paper-Scissors games in individuals with schizophrenia. Psychiatry Research. 207, 158–163.

Brysbaert, M. (2019) How many participants do we have to include in properly powered experiments? A tutorial of power analysis with reference tables. Journal of Cognition, 16, 1-38.

Damaso, K., Williams, P. & Heathcote, A. (2020). Evidence for different types of errors being associated with different types of post-error changes. Psychonomic Bulletin and Review, 27, 435–440.

Dyson, B. J., Wilbiks, J. M. P., Sandhu, R., Papanicolaou, G. & Lintag, J. (2016). Negative outcomes evoke cyclic irrational decisions in Rock, Paper, Scissors. Scientific Reports, 6: 20479.

Dyson, B. J., Sundvall, J., Forder, L. & Douglas, S. (2018). Failure generates impulsivity only when outcomes cannot be controlled. Journal of Experimental Psychology: Human Perception and Performance, 44, 1483-1487.

Dyson, B. J. (2019). Behavioural isomorphism, cognitive economy and recursive thought in non-transitive game strategy. Games, 10: 32.

Dyson, B. J., Musgrave, C. Rowe, C. & Sandhur, R. (2020). Behavioural and neural interactions between objective and subjective performance in a Matching Pennies game. International Journal of Psychophysiology, 147, 128-136. 

Dyson, B. J. (2021). Variability in competitive decision-making speed and quality against exploiting and exploitable opponents. Scientific Reports, 11: 2859.

Faul, F., Erdfelder, E., Lang, A.-G., & Buchner, A. (2007). G*Power 3: A flexible statistical power analysis program for the social, behavioral, and biomedical sciences. Behavior Research Methods, 39, 175-191.

Kahneman, D. & Tversky, A. (1979). Prospect Theory: An analysis of decision under risk. Econometrica, 47, 263–291.

Pulford, B. D., Colman, A. M. & Loomes, G. (2018). Incentive magnitude effects in experimental games: Bigger is not necessarily better. Games, 9: 4. 

Stöttinger, E.; Filipowicz, A.; Danckert, J.; Anderson, B. (2014). The effects of prior learned strategies on updating an opponent’s strategy in the Rock, Paper, Scissors game. Cognitive Science, 38, 1482–1492.

Wang, Z.; Xu, B. (2014). Incentive and stability in the Rock-Paper-Scissors game: An experimental investigation. arXiv:1407.1170.

---

## [Decision Letter · Decision Letter 1]

13 Jun 2022

Assessing behavioural profiles following neutral, positive and negative feedback

PONE-D-22-06419R1

Dear Dr. Dyson,

We’re pleased to inform you that your manuscript has been judged scientifically suitable for publication and will be formally accepted for publication once it meets all outstanding technical requirements.

Kind regards,

Alberto Antonioni, PhD

Academic Editor

PLOS ONE

Additional Editor Comments (optional):

Considering both reviewers' positive evaluation, the work can be accepted after a very minor revision without going through any additional review process. The authors can just include constructive suggestions from Reviewer 2 in the final version of their manuscript.

Reviewers' comments:

Reviewer's Responses to Questions

**Comments to the Author**

1. If the authors have adequately addressed your comments raised in a previous round of review and you feel that this manuscript is now acceptable for publication, you may indicate that here to bypass the “Comments to the Author” section, enter your conflict of interest statement in the “Confidential to Editor” section, and submit your "Accept" recommendation.

Reviewer #1: All comments have been addressed

Reviewer #2: (No Response)

2. Is the manuscript technically sound, and do the data support the conclusions?

Reviewer #1: Yes

Reviewer #2: Yes

3. Has the statistical analysis been performed appropriately and rigorously? 

Reviewer #1: Yes

Reviewer #2: Yes

4. Have the authors made all data underlying the findings in their manuscript fully available?

Reviewer #1: Yes

Reviewer #2: Yes

5. Is the manuscript presented in an intelligible fashion and written in standard English?

Reviewer #1: Yes

Reviewer #2: Yes

6. Review Comments to the Author

Reviewer #1: Thank you very much for your responses. My questions have been answered adequately and I don't not have any further comments. Congratulations to this interesting paper.

Reviewer #2: Thank you for the opportunity to review the revised manuscript. I appreciate the authors’ detailed and thoughtful responses, and most of my comments have been addressed very satisfactorily. I have only a few relatively minor comments left for the revised manuscript.

1. P9. The power analysis is based on the difference between draw (0)-shift and loss (-1)-shift observed in two previous studies. However, this particular comparison does not seem to be the main focus of the paper. For instance, none of the four predictions listed for Experiment 1 (on p.8-9) is about this particular comparison. Given the large number of tests, I think it would be helpful if the authors could clearly state which analyses are theoretically most relevant, and base the power consideration on the theoretically informative effects. Since the sample sizes are known, sensitivity analysis rather than a priori power analysis may be more appropriate here.

2. P5. “However, outcome frequency does not provide a complete account of post-loss slowing since post-loss slowing is intact when positive and negative outcomes are experienced to the same degree (eg., Eben et al., 2020)”. Eben et al. found post-loss speeding, rather than slowing.

3. P5. “A second metric to ascertain the subjective interpretation of draws is the flexibility of responding following outcomes.” Perhaps the authors forgot to change this sentence in the manuscript?

7. PLOS authors have the option to publish the peer review history of their article (what does this mean?). If published, this will include your full peer review and any attached files.

Reviewer #1: **Yes: **Charlotte Eben

Reviewer #2: **Yes: **Zhang Chen

---

## [Editor Report · Acceptance letter]

25 Jun 2022

PONE-D-22-06419R1 

Assessing behavioural profiles following neutral, positive and negative feedback 

Dear Dr. Dyson:

I'm pleased to inform you that your manuscript has been deemed suitable for publication in PLOS ONE. Congratulations! Your manuscript is now with our production department. 

Kind regards, 

on behalf of

Dr. Alberto Antonioni 

Academic Editor

PLOS ONE